# Social behavioural adaptation in Autism

**Baudouin Forgeot d'Arc**[1,2], **Marie Devaine**[3,4], **Jean Daunizeau**[3,4]*

**1** Département de Psychiatrie, Université de Montréal, Montréal, Canada, **2** Centre Intégré Universitaire de Santé et Services Sociaux de Nord-de-l'Île-de-Montréal, Montréal, Canada, **3** Université Pierre et Marie Curie, Paris, France, **4** Institut du Cerveau et de la Moelle épinière, INSERM UMRS 1127, Paris, France

* jean.daunizeau@gmail.com

## Abstract

Autism is still diagnosed on the basis of subjective assessments of elusive notions such as interpersonal contact and social reciprocity. We propose to decompose reciprocal social interactions in their basic computational constituents. Specifically, we test the assumption that autistic individuals disregard information regarding the stakes of social interactions when adapting to others. We compared 24 adult autistic participants to 24 neurotypical (NT) participants engaging in a repeated dyadic competitive game against artificial agents with calibrated reciprocal adaptation capabilities. Critically, participants were framed to believe either that they were competing against somebody else or that they were playing a gambling game. Only the NT participants did alter their adaptation strategy when they held information regarding others' competitive incentives, in which case they outperformed the AS group. Computational analyses of trial-by-trial choice sequences show that the behavioural repertoire of autistic people exhibits subnormal flexibility and mentalizing sophistication, especially when information regarding opponents' incentives was available. These two computational phenotypes yield 79% diagnosis classification accuracy and explain 62% of the severity of social symptoms in autistic participants. Such computational decomposition of the autistic social phenotype may prove relevant for drawing novel diagnostic boundaries and guiding individualized clinical interventions in autism.

**Data Availability Statement:** We note that we have already made our entire data analysis code entirely available as part of an open-science collaborative project (see https://mbb-team.github.io/VBA-toolbox/). In addition, our raw data is accessible

## Author summary

Autism or AS is mostly characterized by impairments in a very specific yet intricate skill set, namely: social intelligence. In this work, we focus on "social reciprocity", i.e. the continuous adaptation of one's behaviour that both moulds and appropriately responds to others' behaviour. Our working hypothesis is that social reciprocity deficits in people with AS derive from a basic inability to tune one's adaptation strategy to contextual knowledge about the stakes of social interactions (e.g., others' cooperative or competitive incentives). We ask participants to engage in simple interactive games with AI agents that are endowed with calibrated reciprocal adaptation capabilities. Critically, participants are framed to believe either that they are competing against somebody else (social framing) or that they are playing a gambling game (non-social framing). Only in the social condition do participants know about the (competitive) incentives of their opponents.

online at the following address: https://owncloud.
icm-institute.org/index.php/s/TsguzHSdgCAQPAL

**Funding:** BFA acknowledges support from
"Fondation Les Petits Trésors de l'Hôpital Rivière
des Prairies" and Fonds de Recherche en Santé du
Québec (FRQS). MD and JD have nothing to
declare. The funders had no role in study design,
data collection and analysis, decision to publish, or
preparation of the manuscript.

**Competing interests:** The authors have declared
that no competing interests exist.

Computational analyses of action sequences in the games show that, contrary to healthy
controls, people with AS do not change their strategy according to whether they hold
information regarding their opponents' incentives or not. In addition, these analyses yield
79% diagnosis out-of-sample classification accuracy (AS versus controls) and predict 62%
of the severity of social symptoms in people with AS. This demonstrates the feasibility of
AI-based quantitative assessments of social cognition and its deficits.

## Introduction

Autism spectrum (AS, or ASD in DSM-5- American Psychiatric Association, 2013; Kenny
et al., 2016) is a highly heterogeneous condition defined by altered reciprocal social interaction
and inflexible patterns of behavior. Despite refinement of diagnostic tools in the last decades,
standardized clinical assessments have limited reliability regarding milder forms of autism
seen in adults and adolescent: we still lack a solid test for autism [1]. In turn, the clinical identi-
fication of autism relies on sociopsychological constructs such as *interpersonal contact* and *rec-
iprocity*, which remain elusive and beyond the reach of objective measurement [2,3]. This
work evaluates the clinical relevance of a computational decomposition of the latter notion,
relying on the quantitative assessment of adaptation strategies in the context of simple dyadic
games.

Most recent neurocognitive work on autism, including computational modelling
approaches, offers an excellent mechanistic account of general perceptual and/or cognitive def-
icits [4–9]. They, however, cannot explain the specific issues autistic people face with social
interactions [10]. Instead, the latter are typically viewed as resulting from an underlying
impairment in Theory of Mind or ToM [11,12], i.e. the ability to understand others' covert
mental states. ToM impairments have been repeatedly evidenced in autistic children using
tests of, e.g., false belief understanding [13–15], sarcasm/irony detection [16,17] or moral eval-
uation [18,19]. However, these tests yield quite unreliable results and have poor psychometric
properties in older individuals [20], including ceiling effects in adolescents and adults [21,22].
This is why, although theoretically relevant to autism, quantitative tests of ToM has had only
limited impact on diagnosis or intervention to date [23].

These mixed results call for a refinement of the "mind blindness" theory of social deficits in
autism [24,25]. In line with recent pleas for "second-person"—i.e. interactionist—approaches
to social cognition [26–29], we propose to reconsider how sociocognitive skills such as ToM
may contribute to *reciprocity*. Reciprocity is a feature of ecological social interactions, the typi-
cal intricacy of which overwhelms autistic people. Not only may subtle variations in social sig-
nals (e.g., facial expressions, speech prosody, etc . . .) reflect profoundly different mental states,
but the stakes of social exchanges may be dynamic, partially implicit, multiple and even con-
flicting (e.g., impose a deal and induce sympathy). In this context, we define reciprocity as the
continuous adaptation of one's behaviour that both moulds and appropriately responds to oth-
ers' behaviour [2,30]. Our working assumption is two-fold. First, we reason that reciprocity
relies on the ability to tune one's adaptation strategy to contextual knowledge about the stakes
of social interactions (e.g., others' cooperative or competitive incentives). In contrast to neuro-
typic controls [31], autistic people may thus not benefit from information regarding others'
incentives when adapting to them. Second, reciprocity may be decomposed into basic (social
and non-social) computational components. Arguably, it should improve with the ease with
which one switches between different cognitive modes and/or behavioural strategies, which
we term *flexibility*. Recent theoretical [32] and empirical [33] work on the evolution of

mentalizing shows that it also critically relies upon *ToM sophistication*, as proxied by the depth of recursive beliefs (as in "I believe that you believe that I believe . . ."). ToM sophistication and flexibility thus provide a minimal computational basis for decomposing reciprocity, which should explain the severity of social symptoms in autism.

We test these assumptions using simple repeated dyadic games, whereby participants play against learning machines endowed with artificial ToM of calibrated sophistication (Baker et al., 2011; Devaine et al., 2014b; Yoshida et al., 2008). To win, participants' must learn to anticipate their opponent's next choice and/or try to influence it. Critically, participants are not told about the algorithmic nature of their opponents. Rather, we have them believe either that they were competing against somebody else (social framing) or that they were playing a gambling game (non-social framing). The objective information available to the participants on each trial is the same for both conditions (actions and feedbacks). However, only in the social condition do participants hold information regarding their opponent's competitive incentives. Critical here is the notion that people may engage the game equipped with a behavioural repertoire composed of many *adaptation strategies*. In appropriate experimental contexts (in particular: dyadic games), these can be disclosed from computational analyses of trial-by-trial choice sequences. One can then measure and compare the computational properties of people's adaptation repertoire, in particular: its ToM-sophistication and its flexibility [31]. In what follows, we refer to these as "computational phenotypes" of social reciprocity. As we will see, they provide a quantitative insight into the specificity of the autistic social phenotype.

## Results

We asked 24 adult participants with ASD and 24 control participants to play repeated dyadic games against artificial "mentalizing" opponents, which differ in their ToM sophistication (hereafter: *k-ToM* agents, see below). In total, each participant played 4x2x2 = 16 games (4 opponent types, 2 framing conditions, 2 repetitions), where each game consisted in 60 successive trials. To succeed, subjects had to anticipate and predict the behaviour of their opponent, who hid himself in one out of two possible locations at each trial (see Fig 1 below).

Opponents either followed a predetermined pseudo-random sequence with a 65% bias for one hand (*RB*), or were designed to deceive the participants from learned anticipations of their behaviour (*0-ToM*, *1-ToM* and *2-ToM*). The difference between *k-ToM* opponents lies in how they learn from the past history of participants' actions, where *k* refers to their calibrated ToM sophistication. In brief, *0-ToM* does not try to interpret the participants' action sequence in terms of a strategic attempt to win. Rather, it simply assumes that abrupt changes in the participants' behaviour are a priori unlikely. It thus tracks the evolving frequency of participants' actions, and chooses to hide the reward where it predicts the opponent will not seek. It is an extension of "fictitious play" learning [34], which can exploit participants' tendency to repeat their recent actions. In contrast, *1-ToM* is equipped with (limited) artificial mentalizing, i.e. it attributes simple beliefs and desires to participants. More precisely, it assumes that participants' actions originate from the strategic response of a *0-ToM* agent that attempts to predict its own actions. Note that the computational sophistication of artificial mentalizing is not trivial, since *1-ToM* has to explicitly represent and update its (recursive) belief about its opponents' beliefs. Practically speaking, *1-ToM* learning essentially consists in an on-line estimation of *0-ToM*'s parameters (e.g., learning rate and behavioural temperature) given the past history of both players' actions. This makes *1-ToM* a so-called "meta-Bayesian" agent [32,35] that can outwit strategic opponents who do not mentalize when competing in the game (such as *0-ToM*). Although *1-ToM* is mentalizing, it is not capable of dealing with other mentalizing agents. This is the critical difference between *1-ToM* and *2-ToM*. At this point,

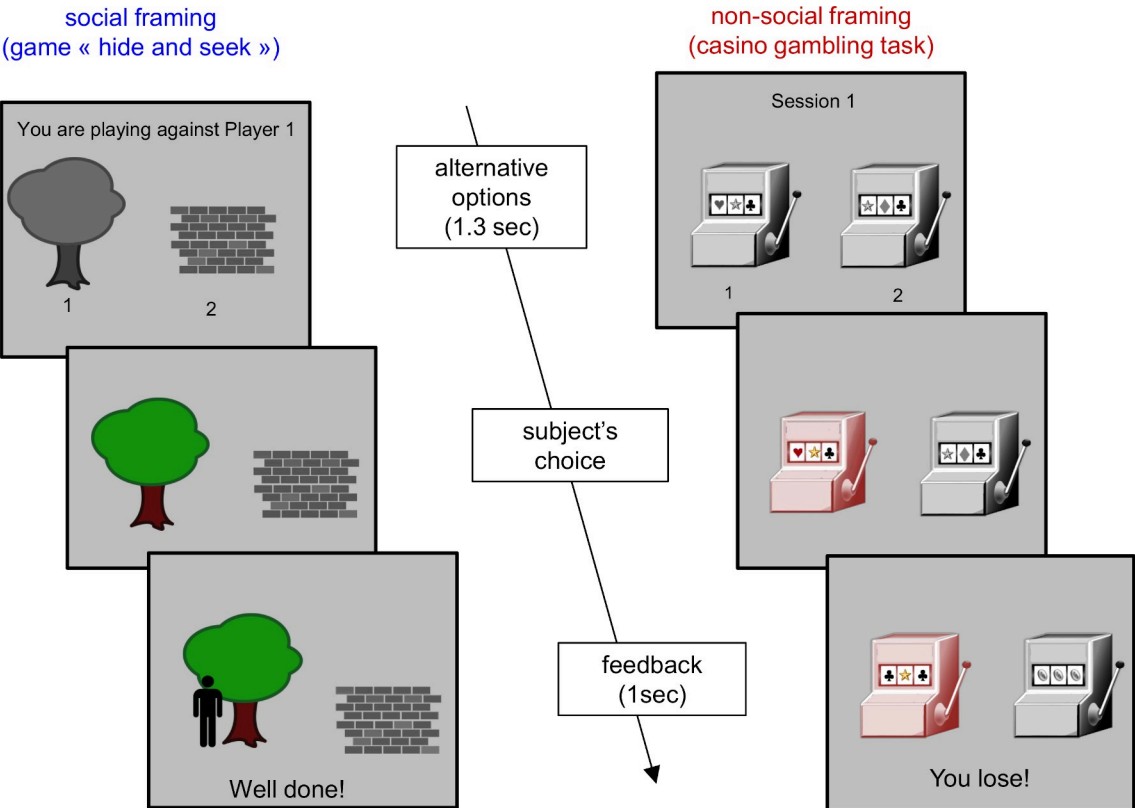

**Fig 1. Experimental protocol.** Left: social framing ("hide-and-seek" game). Right: non-social framing (gambling game). At each trial, participants have 1300 msec to pick one of the two options (social framing: wall or tree, non-social framing: left or right slot machine). Feedback is displayed for 1 sec; and includes the trial outcome (win or loss) and the actual winning option (social framing: character picture, non-social framing: three identical items).

suffices to say that *2-ToM* is an artificial mentalizing agent that can learn to predict how other mentalizing agents (such as *1-ToM*) will behave.

Critically, participants were not cued about opponent conditions. This implies that they had to adapt their behaviour according to their understanding of the history of past actions and outcomes. In addition, except in the control (*RB*) condition, there is no possibility to learn the correct answer from simple reinforcement. This is because *k-ToM* artificial learners exhibit no systematic preference for any particular action. Further details regarding the experimental protocol as well as *k-ToM* artificial agents can be found in the Methods section below.

We first focus on peoples' ability to alter their adaptation strategy as a function of whether or not they hold information about their opponents' competitive incentives. Fig 2 below summarizes the performance results, in terms of the net rate of correct answers in each of 4x2 conditions, for both (NT and AS) groups.

One can see that the performance patterns are markedly different between NT and AS participants. To begin with, the performance of NT participants qualitatively reproduces previous experiments with healthy human adults [31]. In brief, in the non-social framing condition, NT participants eventually lose against artificial mentalizing agents (*1-ToM* and *2-ToM*) whereas they maintain their earnings in the social framing condition. The AS group however, seems to show no effect of the framing manipulation, i.e. their performance pattern across opponents is the same, irrespective of whether they know about their opponent's competitive incentives. Interestingly, they seem to lose against artificial mentalizing agents (as NT controls in the non-

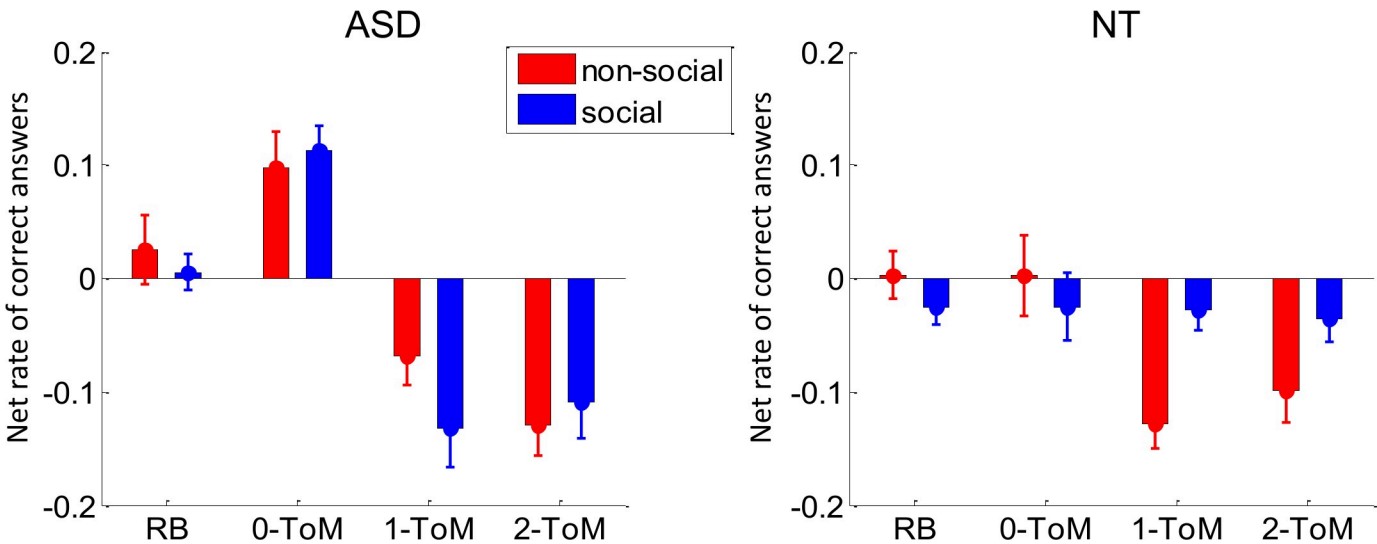

**Fig 2. Behavioural performance results.** Group average net rate of correct answers (y-axis) against the four opponent types (x-axis) for both framing conditions (blue: social, red: non-social) in both AS (left) and control (right) participants. Note: The net rate of correct answers is defined as $(n_c - n_i)/(n_c + n_i)$, where $n_c$ and $n_i$ are the number of correct and incorrect responses, respectively. Hence, it is null when participants perform at chance level (50% accuracy). In this and all subsequent figures, error bars depict the standard error around the mean.

social framing condition), but they outperform NT controls against non-mentalizing learning agents (*0-ToM*). We performed a pooled variance ANOVA to assess the statistical significance of these observations. We found a significant three-way interaction between group (AS vs NT), opponent and framing (F[3,690] = 3.6, p = 0.014, $R^2$ = 1.5%), a significant interaction between group and opponent (F[3,690] = 9.5, p<$10^{-4}$, $R^2$ = 4.0%) and a main effect of opponent (F[3,690] = 33.7, p<$10^{-4}$, $R^2$ = 12.8%). We then looked more closely at the three-way interaction using post-hoc tests. In the NT group, there was a main effect of opponent (F = 4.5, p = 0.004), no main effect of framing (F = 2.6, p = 0.11) but a significant interaction opponent x framing (F = 3.7, p = 0.011). In the AS group, there was a main effect of opponent (F = 38.7, p<$10^{-4}$) but no main effect of framing (F = 0.5, p = 0.46) nor interaction (F = 1.3, p = 0.27). In other terms, only NT participants show the opponent x framing interaction. This is because NT participants perform better in the social than in the non-social framing only against artificial mentalizing agents (p<$10^{-4}$). Now focusing on performances against artificial mentalizing agents, there was a significant interaction between group and framing (p = 0.001). This is because against *1-ToM* and *2-ToM*, NT participants perform significantly better than AS people against artificial mentalizing agents in the social framing (p<$10^{-4}$) but not in the non-social framing (p = 0.65). Besides, AS participants perform significantly better than NT participants against *0-ToM* (p<$10^{-4}$), and this effect does not depend upon the game's framing (p = 0.46).

One of the main differences between NT and AS participants is thus that the latter seem to be insensitive to information regarding their opponents' competitive incentives. This is in fact confirmed by additional analyses showing that (i) performance variations induced by opponent types in different framing conditions are significantly correlated (see section 5 in S1 Text), and (ii) model-free decompositions of their trial-by-trial choice sequences show no effect of framing (see section 6 in S1 Text).

At this point, we asked whether we could classify AS and NT participants based upon their performance patterns in the task. Averaging performances over repetitions yielded a feature space of 8 dimensions (4 opponent types, 2 framings), which was then fed to a classifier based

upon logistic regression [36]. Test classification accuracy was evaluated using a simple leave-one-out cross-validation scheme. The classifier achieved 73% of correct out-of-sample classifications, which is statistically better than chance (p = 0.001). This will serve as a reference point for evaluating the added-value of computational phenotypes.

We now ask whether differences in computational phenotypes such as ToM sophistication and flexibility predict social deficits. We considered a set of eight distinct adaptation strategies that constitute peoples' potential behavioural repertoire. Somewhere at the end of the sophistication spectrum lie social adaptation strategies that derive from recursive ToM [31,37,38]. We also considered adaptation strategies that take simpler forms, ranging from mundane heuristics, to trial-and-error learning, to cognitive shortcuts of ToM that simply care about tracking others' overt reaction to one's own actions [39]. Each of these adaptation strategies corresponds to a formal learning/decision model that provides a probabilistic prediction of observed peoples' trial-by-trial choice sequences. We then performed a subject-specific bayesian model comparison of these models. Note that, in contrast to the NT group which shows strong inter-individual variability in terms of behavioural strategies, trial-by-trial choice sequences of most AS players, in both framing conditions, are captured by a single model, namely: "influence learning" (see section 7 in the Supplementary Text). We then evaluated both the flexibility ($\hat{f}$, rate of strategy switching) and the ToM-sophistication ($\hat{k}$, recursive depth of beliefs) of peoples' behavioural repertoire. We refer the interested reader to the Methods section.

We first asked whether control and AS participants would show differences in their repertoire's ToM-sophistication. Fig 3 below shows the repertoire's ToM-sophistication $\hat{k}$ averaged across repetitions, across opponent conditions and across participants, for each group and for both framing conditions.

A simple ANOVA shows no evidence for an interaction between group and framing (F [1,46] = 0.6, p = 0.42, $R^2$ = 1.4%), no main effect of framing (F[1,46] = 1.8, p = 0.18, $R^2$ = 3.8%), but a significant group effect (t[46] = 1.9, p = 0.03, $R^2$ = 7.3%). Post-hoc tests show that this group difference is mostly driven by the social framing condition (t[46] = 1.9, p = 0.03, $R^2$ = 7.5%), whereas there is no significant group difference in the non-social condition (t[46] = 1.1,

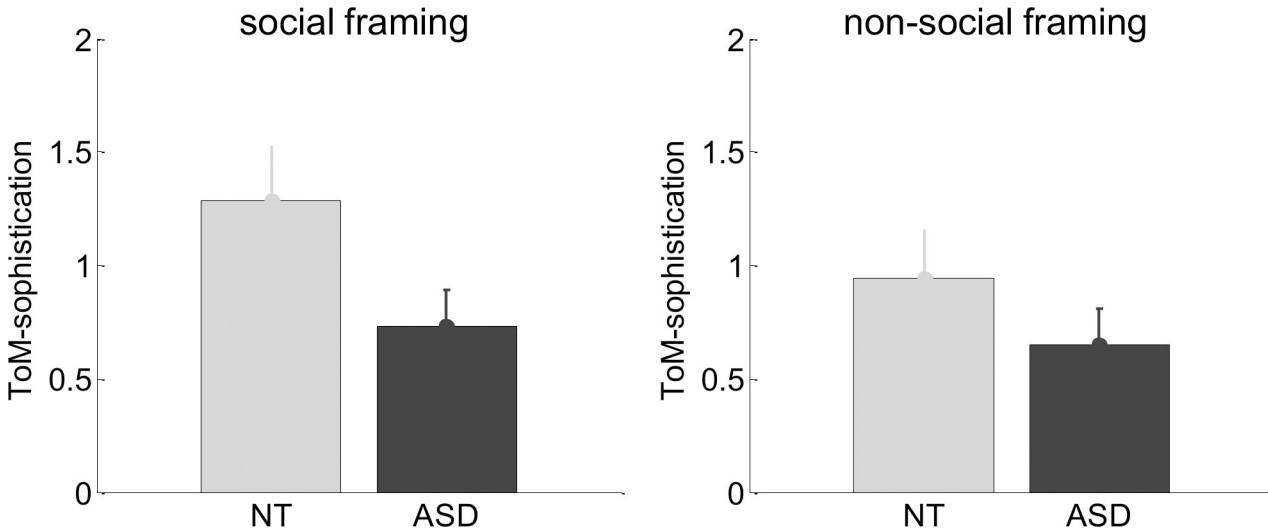

**Fig 3. Model-based analysis of trial-by-trial choice sequences: ToM sophistication scores.** ToM sophistication scores are shown as a function of framing conditions (left: social, right: non-social) for both control (gray) and AS participants (back).

$p = 0.13$, $R^2 = 2.7\%$). In other words, only in the social framing do control participants exhibit higher ToM-sophistication than AS participants.

We then investigated whether control and AS participants show differences in their repertoire's flexibility. Fig 4 below shows the repertoire's flexibility $\hat{f}$, both across framings and across repetitions. The former measures peoples' tendency to change their adaptation strategy in response to information regarding others' incentives. The latter can be thought of as a base rate of strategy switching, across identical situations. Note that, when evaluating flexibility between repetitions separately in both framing conditions, only in the NT group is it significantly increased when participants know about others' incentives (see section 8 in S1 Text). Here again, there is no significant interaction between group and flexibility type ($F[1,46] = 0.55$, $p = 0.46$, $R^2 = 1.2\%$), but there is a significant main effect of flexibility type ($F[1,46] = 5.54$, $p = 0.02$, $R^2 = 10.7\%$) and a main effect of group ($t[46] = 3.4$, $p = 0.001$, $R^2 = 20.4$). Post-hoc tests show that this group difference in repertoire's flexibility is strong both across framings ($t[46] = 3.4$, $p = 0.001$, $R^2 = 20.7\%$) and across repetitions ($t[46] = 2.8$, $p = 0.004$, $R^2 = 14.4\%$). Also, AS participants show no "flexibility gap", i.e. no difference between flexibility across framings and flexibility across conditions ($p = 0.26$). This contrasts with control participants, who exhibit a significant flexibility gap ($p = 0.03$).

If only, this computational analysis confirms that AS participants exhibit a distinct pattern of social computational phenotypes (when compared with NT controls). But do the latter provide diagnosis-relevant information, above and beyond performance scores in the task? When augmenting the previous classifier with social computational phenotypes, classification accuracy reaches 79% of correct out-of-sample classifications ($p < 10^{-4}$). This matches the diagnosis reliability of trained psychologists, as measured in terms of the inter-rater agreement rate in the use of the standard Autism Diagnosis Observation Schedule [40]. Note that the probability that a (yet unseen) individual will be better classified with than without computational phenotypes is 0.79, and that inter-individual variability in flexibility does not correlate with ToM sophistication (see section 9 in the Supplementary Text). This is important, since it means that all computational phenotypes bring additional, diagnosis-relevant, information.

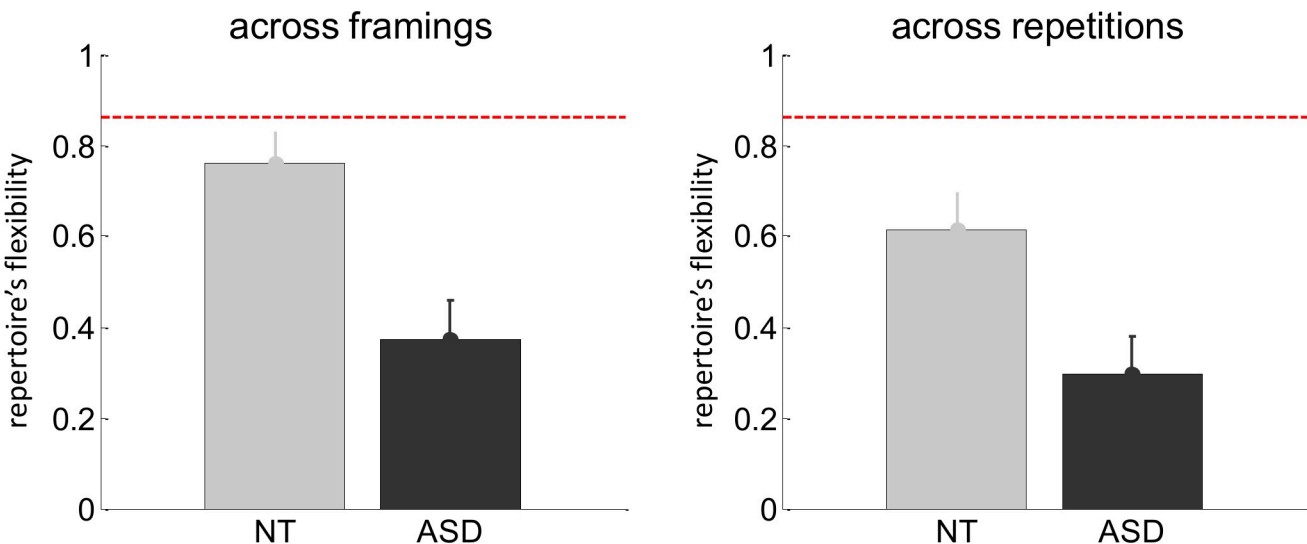

**Fig 4. Model-based analysis of trial-by-trial choice sequences: Repertoire's flexibility.** The repertoire's flexibility is shown across framing conditions (left) and across repetitions (right) for both control (gray) and AS participants (back).

Finally, we asked whether we could predict, from estimated computational phenotypes, inter-individual variations in symptom severity among AS participants. More precisely, we focused on the 'social' and 'stereotyped behavior' subscores of the ADOS scale, which quantify social and non-social deficits, respectively. We found that inter-individual differences in computational phenotypes predict social deficits with high accuracy (F[4,15] = 6.1, p = 0.004, $R^2$ = 62.1%), but do not predict non-social deficits (F[4,15] = 1.5, p = 0.25, $R^2$ = 28.8%). Post-hoc univariate tests actually show that social deficits significantly decrease with ToM sophistication improvement $\Delta \hat{k} = \hat{k}_{soc} - \hat{k}_{NS}$ (t[18] = -1.8, p = 0.04, $R^2$ = 15.9%) and with the flexibility gap $\Delta \hat{f} = \hat{f}_{framing} - \hat{f}_{repetitions}$ (t[18] = -2.6, p = 0.009, $R^2$ = 27.5%). This concludes our computational decomposition of social reciprocity and its alteration in AS.

## Discussion

Maybe the most striking result of our work is that autistic people are insensitive to the task framing, i.e. they do not adjust their adaptation strategy in response to information about others' incentives. Recall that we demonstrated this in three different ways: (i) AS participants show no difference between performance or ToM-sophistication scores between framing conditions (cf. Fig 2), (ii) model-free decompositions of their trial-by-trial choice sequences show no effect of framing (see section 6 in S1 Text), and (iii) their behavioural repertoire exhibits very low flexibility across framing conditions (cf. Fig 4). Importantly, participants' debriefing showed that the framing manipulation was similarly credible in both groups of subjects (see section 2 in S1 Text). In line with social motivational theories of autism [41], one may argue that, in contrast to control participants, AS participants may not have been interested enough to invest the cognitive effort required for improving their performance in the social framing condition. Such global motivational and/or attentional interpretations are unlikely however, because AS participants actually outperform controls against *0-ToM* in the social framing condition. In addition, financial incentive manipulations have no effect on performance in the game, for both AS and NT groups. This is despite the fact that both groups are consistently and equally sensitive to monetary incentives in the context of cognitive control tasks (see section 3 in S1 Text). Taken together, our results support the idea that adults with AS are not unwilling, but rather unable to exploit knowledge about the stakes of social interactions when adapting to others. However, it remains possible that social motivation might account for other aspects of autism, for example in altering the normal scaffolding of social cognition during development.

Of particular interest is the finding that autistics outperform controls in certain conditions of the game. In particular: they win against non-mentalizing learning opponents, irrespective of the task framing. Given that control participants merely achieve null earnings in the same condition, this result is a striking demonstration of the efficiency of autistics' behavioural strategy. Although strengths and peaks of ability have been reported since the first descriptions of autism as core features, they have been largely ignored in the more recent scientific literature, with few exceptions (Ostrolenk, Forgeot d'Arc, Jelenic, Samson, & Mottron, 2017). In fact, a possible explanation for such success is that non-mentalizing agents are somehow more "autistic", i.e. more similar to patients' expectations (see below). This is reminiscent to the so-called "social interaction mismatch" hypothesis, which suggests that autistic persons find it easier to relate to other autistic persons [42,43]. In any case, future studies including measures of everyday functioning might test whether such performance peaks in the task relate to autistic strengths in real-life situations.

One may ask whether the performance pattern we report here may not be due to the fact that AS individuals are typically slower than neurotypic people. This would be because in the

task, participants have only 1.3 second to respond, which would potentially be too short for AS individuals to reach the correct decision. The ensuing "behindhand errors" could then confound analyses of performance data. In particular, this would explain performance differences in situations where strategic thinking is (in principle) most needed, i.e. against mentalizing agents in the social framing condition. We find that this is an unlikely confound however, because the pattern of behindhand responses (i.e. responses whose RT reaches the decision time limit) and performance are globally inconsistent with each other (cf. section 4 of the S1 Text). Nevertheless, we acknowledge that dual analyses of concurrent performance/RT data (such as those based upon accumulation-to-bound models that generate both choice and RT data) may indeed provide insights into the neural implementational processes underlying our behavioural results. This may be addressed in future work.

One may also question the nature of the social cognitive processes that the task assesses. Although our original intention was to address the elusive notions of *contact* and *reciprocity*, the task itself falls short of a few important features of real-life social interactions. For example, it ignores the diversity of social signals (e.g., verbal/body language, facial expressions) and modulatory factors (e.g., in-group/out-group context, familiarity) that are relevant for establishing contact with others. It also does not involve changes in others' intentions (e.g., competitive/cooperative) and/or attitudes (e.g., friendly/aggressive, dominant/submissive), which would be necessary to assess certain aspects of social reciprocity. Instead, it focuses on peoples' ability to respond and/or influence others' actions in a simplified competitive setting. Clearly, this cannot account for the breadth of social cognitive processes that underlie contact and reciprocity. One might even think that task performance may not load very heavily on social cognitive processes, when compared with other instrumental processes (e.g., working memory or reasoning). This is unlikely, however, given that we have shown, in a very large online population sample, that a very small amount of inter-individual variance in the game's performance can be predicted from cognitive control skills [44]. In any case, we think the simplicity of our task design also has its virtues. This is because it eventually enables us to construct a non-social control condition that is matched with the social condition in terms of goal-relevant information (cf. trial-by-trial feedbacks). This turns out to be critical to discriminate between AS and NT participants.

Now is our approach really useful for clinical purposes? That it can achieve 79% of accuracy in diagnostic prediction is only relevant for comparing this test with other tests of the same kind, or as a proof-of-concept demonstration. In fact, the long-term goal of approaches of this kind is not to reflect the diagnosis per se (which is irrelevant), but rather to guide clinical decisions. Ideally, a useful approach should reflect a pathological mechanism and predict outcome and/or treatment response. Diagnosis is just a proxy for such prediction, and one has to admit that current psychiatric categorical diagnoses are not quite satisfactory in this regard [45]. In turn, evaluating the clinical utility of our approach would require assessing how it relates to genetic variants, brain metrics, specific outcomes or response to intervention. This is beyond the scope of the current work, but we intend to pursue these issues in forthcoming publications. That our approach predicts 62% of inter-individual variance in social symptoms may be more interesting at first sight. This is because explaining variability beyond categorical diagnosis may be relevant for identifying clinical subcategories. But here again, establishing the clinical utility of such findings can only be done on the basis of, e.g. treatment outcome prediction. In addition, such significant but modest explanatory power is a reminder that social symptoms in AS are not solely due to mentalizing deficits. For example, they could be driven by some other issues in social cognition, including, but not limited to, social anxiety [46] or the misperception/misunderstanding of social norms [47,48]. Evaluating the relative contribution of

these processes to social symptoms is clearly a promising research avenue for the computational psychiatric approach to AS.

Let us now discuss the main qualitative difference between adaptation strategies in people with and without autism. If anything, the adaptation strategies of NT control participants exhibit strong intra- and inter-individual variability. In contrast, trial-by-trial choice sequences of most AS players, in both framing conditions, are captured by a single model, namely: "influence learning" [39]. From a computational standpoint, this model possesses broad adaptive fitness because it essentially is a generic way of dealing with environments that react to one's actions [33]. In other words, influence learning can be seen as an all-purpose cognitive toolkit that would be expected to perform well in a wide range of contexts, excluding competitive interactions with mentalizing agents (cf. pattern of performances against *RB*, *0-ToM*, *1-ToM* and *2-ToM* in section 9 in the S1 Text). Note that this explains why AS participants perform better than NT controls against non-mentalizing agents (in both framing conditions), and why they show worse performance against mentalizing agents (in the social framing condition). That they rely on influence learning in both framings also explains their lower flexibility score, as well as the absence of a framing effect on raw performance. Strictly speaking, an agent capable of influence learning is thus not "mind blind", but it cannot adjust its behavioural strategy to the intentions of mentalizing agents. In other words, even if equipped with a sophisticated perceptual apparatus (that would enable the recognition of ecological social signals), an influence learner would show limited social reciprocity. Reliance on this -or similar- adaptation strategy thus provides a computational explanation for an important aspect of the autistic social phenotype, which would not depend upon motivational factors and/or cognizance of the social context.

Obviously, our experimental claim does not go as far as to assert that the behavioural repertoire of autistic people is generally limited to influence learning. Nevertheless, it clearly exhibits subnormal flexibility, which corroborates previous reports of executive dysfunction in autism [5,49,50]. Note that, together with ToM sophistication, our computational measure of flexibility contributes to predict social symptoms and AS diagnosis. It does not, however, relate to the ADOS' index of repetitive behaviours. This may be because repetitive behaviours in autism tend to decrease with age [51] and might not be consistently accessible through direct observation during the administration of the ADOS [3]. Interestingly, only in the NT group is flexibility (between repetitions) significantly higher in the social than in the non-social framing condition (see section 7 in S1 Text). And inter-individual variability in flexibility does not correlate with ToM sophistication (see section 8 in the Supplementary Text). This suggests that impairments in flexibility may contribute to social deficits, independently of mentalizing skills [52–54]. This is important, because inter-individual differences in flexibility and ToM sophistication may separately contribute to diversity in the autism spectrum. Thus, these computational phenotypes may serve to draw novel diagnostic boundaries and guide individualized clinical interventions [6].

## Methods

### Ethics statement

Behavioural assessments were performed in accordance with institutional ethical guidelines, which comply with the guidelines of the declaration of Helsinki. The research protocol was approved by the Ethical Committee of the Hôpital Rivière-des-Prairies, Montréal, where the tests were performed.

### Experimental methods

Participants: n = 24 adults with ASD without mental nor language deficiency and n = 24 NT control subjects participated in the study. All subjects were French speakers (Québec), and

both groups were matched in terms of gender balance (AS: 21 males, NT: 21 males), age (AS: 25.5 y.o. ± 5.7; NT: 27.9 y.o. ± 8.6) and IQ (AS: 104 ± 17; NT: 106 ± 14). AS participants were assessed with ADOS-G and met DSM-5 criteria for ASD. NT participants went through a semi-structured interview to screen for any psychiatric treatment history, learning disorders, personal or family history (2 degrees) for mood disorder, ASD or schizophrenia. No included participant reported strong depressive symptoms (Beck depression Inventory score<20). All participants gave their informed consent, were fully debriefed at the end of the experiment, and received a financial compensation for their participation.

The behavioural task consists of a computerized game (60 trials each) with two framing conditions. In the *social* condition, the task was framed as an online competitive game with someone else. In the *non-social* condition, it was framed as a gambling game. In fact, both games were played against four different learning algorithms with different artificial mentalizing sophistication (ranging from a random sequence with a bias to so-called *2-ToM* agents: see below). Note that, on top of framing and opponent factors, we also varied the financial payoff attached to a correct answer in the games. More precisely, the maximal payoff that participants could earn over one game session was either 10$ (high reward condition) or 1 cent (low reward condition). This manipulation, however, did not induce any effect (cf. section 3 of the S1 Text). In what follows, we thus refer to this experimental factor as a repetition of the task conditions. At each trial, subjects had 1300 ms to make a binary choice (the place to hide or the slot machine to try), which was fed to the learning algorithms to compute online predictions of the participant's action at the next trial. In total, each participant performed 2×4×2 = 16 games (2 framings, 4 opponent types, 2 repetitions) in a pseudo-randomized order. We refer the interested reader to the S1 Text for more details regarding the experimental protocol.

## Computational modelling of adaptation strategies

In this section, we give a brief overview of the set of candidate learning/decision models, with a particular emphasis on *k-ToM* models (because these are also used as on-line algorithms during the experimental phase). We will consider repeated dyadic (two-players) games, in which only two actions are available for each player (the participant and his opponent). Hereafter, the action of a given agent (resp., his opponent) is denoted by $a^{self}$ (resp., $a^{op}$). By convention, actions $a^{op}$ and $a^{self}$ take binary values encoding the first ($a = 1$) and the second ($a = 0$) available options. A game is defined in terms of its payoff table, whose entries are the player-specific utility $U(a^{self}, a^{op})$ of any combination of players' actions at each trial. In particular, competitive social interactions simply reduce to anti-symmetric players' payoff tables (see Table 1 below).

According to Bayesian decision theory, agents aim at maximising expected payoff $V = E[U(a^{self}, a^{op})]$, where the expectation is defined in relation to the agent's uncertain predictions about his opponent's next move. This implies that the form of the decision policy is the same for all agents, irrespective of their ToM sophistication. Here, we consider that choices may exhibit small deviations from the rational decision rule, i.e. we assume agents employ the so-

**Table 1. Competitive payoff table (hider's payoff, seeker's payoff).** Participants play the role of the seeker, the opponent is the hider.

| Seeker \ Hider | $a = 1$ | $a = 0$ |
|---|---|---|
| $a = 1$ | 1,0 | 0,1 |
| $a = 0$ | 0,1 | 1,0 |

called "softmax" probabilistic policy:

$$P(a^{self} = 1) = \frac{1}{1 + \exp\left(-\frac{\Delta V}{\beta}\right)} \tag{1}$$

where $P(a^{self} = 1)$ is the probability that the agent chooses the action $a^{self} = 1$, $\Delta V$ is the expected payoff difference (between actions $a^{self} = 1$ and $a^{self} = 0$), and $\beta$ is the so-called behavioural "temperature" (which controls the magnitude of deviations from rationality). The sigmoidal form of Eq 1 simply says that the probability of choosing the action $a^{self} = 1$ increases with the expected payoff difference $\Delta V$, which is given by:

$$\Delta V = p^{op}(U(1,1) - U(0,1)) + (1 - p^{op})(U(1,0) - U(0,0))$$
$$= 2p^{op} - 1 \tag{2}$$

where $p^{op}$ is the probability that the opponent will choose the action $a^{op} = 1$, and the second line derives from inserting the above payoff matrix (Table 1). In brief, Eq 2 simply says that participants are rewarded for correctly guessing where their opponent is hiding.

Let us now summarize the mathematical derivation of *k-ToM* models, which essentially differ in how they estimate $p^{op}$ from the repeated observation of their opponent's behaviour. We will see that *k* indexes a specific form of ToM sophistication, namely: the recursive depth of learners' beliefs (as in "I believe that you believe that I believe . . ."). Note that *k-ToM*'s learning rule can be obtained recursively, starting with *0-ToM* [32].

By convention, a *0-ToM* agent does not attribute mental states to his opponent, but rather tracks his overt behavioural tendency without mentalizing. More precisely, *0-ToM* agents simply assume that their opponents choose the action $a^{op} = 1$ with probability $p^{op} = s(x_t)$, where the unknown log-odds $x_t$ varies across trials *t* with a certain volatility $\sigma^0$ (and *s* is the sigmoid function). Observing his opponent's choices gives *0-ToM* information about the hidden state *x*, which can be updated trial after trial using Bayes rule, as follows:

$$\mu_t^0 \approx \mu_{t-1}^0 + \Sigma_t^0(a_t^{op} - s(\mu_{t-1}^0))$$
$$\Sigma_t^0 \approx \frac{1}{\frac{1}{\Sigma_{t-1}^0 + \sigma^0} + s(\mu_{t-1}^0)(1 - s(\mu_{t-1}^0))} \tag{3}$$

where $\mu_t^0$ (resp. $\Sigma_t^0$) is the approximate mean (resp. variance) of *0-ToM*'s posterior distribution $p(x_t^0 | a_{1:t}^{op})$. Inserting $\hat{p}_{t+1}^{op} = E[s(x_{t+1}) | a_{1:t}^{op}]$ into Eq 1 now yields *0-ToM*'s decision rule. Here, the effective learning rate is the subjective uncertainty $\Sigma^0$, which is controlled by the volatility $\sigma^0$. At the limit $\sigma^0 \rightarrow 0$, Eq 3 converges towards the (stationary) opponent's choice frequency and *0-ToM* essentially reproduce "fictitious play" strategies [34].

*0-ToM*'s learning rule is the starting point for a *1-ToM* agent, who considers that she is facing a *0-ToM* agent. This means that *1-ToM* has to predict *0-ToM*'s next move, given his beliefs and the choices' payoffs. The issue here is that *0-ToM*'s parameters (volatility $\sigma^0$ and exploration temperature $\beta$) are unknown to *1-ToM* and have to be learned, through their non-trivial effect on *0-ToM*'s choices. At trial *t+1*, a *1-ToM* agent predicts that *0-ToM* will chose the action $a^{op} = 1$ with probability $p_{t+1}^{op,0} = s \circ v^0(x_t^0, a_{1:t})$, where the hidden states $x_t^0$ lumps $\sigma^0$ and $\beta$ together and the mapping $v^0$ is derived from inserting *0*-ToM's learning rule (Eq 3) into Eqs 1 and 2. Similarly to *0-ToM* agents, *1-ToM* assumes that the hidden states $x_t^0$ vary across trials with a certain volatility $\sigma^1$, which yields a meta-Bayesian learning rule similar in form to *0-ToM*'s, but relying on first-order meta-beliefs (i.e. beliefs about beliefs). In brief, *1-ToM*

eventually learns how her (*0-ToM*) opponent learns about herself, and acts accordingly (cf. Eqs 1 and 2).

*1-ToM* agents are well equipped to deal with situations of observational learning. However, when it comes to reciprocal social interactions, one may benefit from considering that others are also using ToM. This calls for learning strategies that rely upon higher-order meta-beliefs. By construction, *k-ToM* agents ($k \geq 2$) consider that their opponent is a *$\kappa$-ToM* agent with a lower ToM sophistication level (i.e.: $\kappa < k$). Importantly, the sophistication level $\kappa$ of *k-ToM*'s opponent has to be learned, in addition to the hidden states $x^\kappa$ that control the opponent's learning and decision making. The difficulty for a *k-ToM* agent is that she needs to consider different scenarios: each of her opponent's possible sophistication level $\kappa$ yields a specific probability $p_{t+1}^{op,\kappa} = s \circ v^\kappa(x_t^\kappa, a_{1:t})$ that she will choose action $a^{op} = 1$. The ensuing meta-Bayesian learning rule entails updating *k-ToM*'s uncertain belief about her opponent's sophistication level $\kappa$ and hidden states $x^\kappa$:

$$\lambda_t^{k,\kappa} \approx \left[ \frac{\lambda_{t-1}^{k,\kappa} \, p_t^{op,\kappa}}{\sum_{\kappa' < k} \lambda_{t-1}^{k,\kappa'} \, p_t^{op,\kappa'}} \right]^{a_t^{op}} \left[ \frac{\lambda_{t-1}^{k,\kappa} (1 - p_t^{op,\kappa})}{\sum_{\kappa' < k} \lambda_{t-1}^{k,\kappa'} (1 - p_t^{op,\kappa'})} \right]^{1 - a_t^{op}}$$

$$\mu_t^{k,\kappa} \approx \mu_{t-1}^{k,\kappa} + \lambda_t^\kappa \, \Sigma_t^{k,\kappa} \, W_{t-1}^\kappa (a_t^{op} - s \circ v^\kappa(\mu_{t-1}^{k,\kappa}))$$

$$\Sigma_t^{k,\kappa} \approx [(\Sigma_{t-1}^{k,\kappa} + \sigma^k)^{-1} + s' \circ v^\kappa(\mu_{t-1}^{k,\kappa}) \lambda_t^\kappa \, W_{t-1}^{\kappa \, T} W_{t-1}^\kappa]^{-1}$$

$$(4)$$

where $\lambda_t^{k,\kappa}$ is *k-ToM*'s posterior probability that her opponent is *$\kappa$-ToM*, and $W^\kappa$ is the gradient of $v^\kappa$ with respect to the hidden states $x^\kappa$. Eq 4 also captures *1*-ToM's learning rule, when setting $\lambda_t^{1,0} \triangleq 1$. Note that although the dimensionality of *k-ToM*'s beliefs increases with *k*, *k-ToM* models do not differ in terms of the number of their free parameters. More precisely, *k-ToM*'s learning and decision rules are entirely specified by their prior volatility $\sigma^k$ and behavioural temperature $\beta$.

Formally speaking, only *k-ToM* agents with $k \geq 1$ are mentalizing about others' covert mental states, i.e. represent and update others' beliefs. They can do this because they adopt the intentional stance [55], i.e. they assume that $p^{op}$ is driven by their opponent's hidden beliefs and desires. More precisely, they consider that the opponent is himself a Bayesian agent, whose decision policy $p^{op} = P(a^{op} = 1)$ is formally similar to Eq 1. This makes *k-ToM* meta-Bayesian learners [35] that relies upon recursive belief updating ("I believe that you believe that I believe . . ."). Critically, the recursion depth *k* induces distinct ToM sophistication levels, whose differ in terms of how they react to the history of players' actions in the game.

With the exception of *0-ToM*, we so far only described sophisticated learning models that are capable of (artificial) ToM. But clearly *0-ToM* is not the only way people may learn in social contexts without mentalizing. We thus consider below other adaptation strategies that may populate peoples' behavioural repertoire.

First, let us consider a heuristic learning model, whose sophistication somehow lies in between *0-ToM* and *1-ToM*. In brief, "influence learning" adjusts a *0-ToM*-like learning rule to account for how her own actions may influence her opponent's behaviour [39]:

$$p_{t+1}^{op} = p_t^{op} + \underbrace{\eta(a_t^{op} - p_t^{op})}_{\text{prediction error}} + \underbrace{\lambda p_t^{op}(1 - p_t^{op})(1 - 2a_t^{self} - \beta s^{-1}(p_t^{op}))}_{\text{"influence" adjustment term}}$$

$$(5)$$

where $\eta$ (resp. $\lambda$) controls the relative weight of its prediction error (resp. the "influence" adjustment term). Numerical simulations show that, in a competitive game setting, *Inf* wins

over *0-ToM* but loses against *k-ToM* players with $k \geq 1$. In other terms, although it is in principle able to adapt to reactive environments, *Inf* cannot successfully compete with learners endowed with mentalizing [33].

Second, participants may learn by trial and error, eventually reinforcing the actions that led to a reward. Such adaptation strategy is the essence of classical conditioning, which is typically modelled using reinforcement learning or *RL* [56]. In this perspective, participants would directly learn the value of alternative actions, which bypasses Eq 2. More precisely, an *RL* agent would update the value of the chosen option in proportion to the reward prediction error, as follows:

$$\begin{cases} V_{t+1}^i = V_t^i + \alpha(R_t - V_t^i) & \text{if action } a_t^{self} = i \text{ was chosen} \\ V_{t+1}^i = V_t^i & \text{otherwise} \end{cases} \quad (6)$$

where $R_t = U(a_t^{self}, a_t^{op})$ is the last reward outcome and $\alpha$ is the (unknown) learning rate. At the time of choice, RL agents simply tend to pick the most valuable option (cf. Eq 1).

Third, an even simpler way of adapting one's behaviour in operant contexts such as this one is to repeat one's last choice if it was successful and alternate otherwise. This can be modeled by the following update in action values:

$$\begin{cases} V_{t+1}^i = R_t & \text{if action } a_t^{self} = i \text{ was chosen} \\ V_{t+1}^i = -R_t & \text{otherwise} \end{cases} \quad (7)$$

This strategy is called win-stay/lose-switch (*WS*), and is almost identical to the above *RL* model when the learning rate is $\alpha = 1$. Despite its simplicity, *WS* can be shown to have remarkable adaptive properties [57].

Last, the agent may simply act randomly, which can be modeled by fixing the value difference to zero ($\Delta V = 0$). Although embarrassingly simple, this probabilistic policy eventually prevents one's opponent from controlling one's expected earnings. It thus minimizes the risk of being exploited at the cost of providing chance-level expected earnings. It is the so-called "Nash equilibrium" of our "hide and seek" game. Since we augment this model with a potential bias for one of the two alternative options (as all the above learning models), we refer to it as *biased Nash* or *BN*.

## Empirical estimates of computational phenotypes

Our working hypothesis is that people may not always rely on the same adaptation strategy across different game sessions or conditions. Rather, they select a strategy from among a repertoire, whose flexibility and ToM sophistication define our computational phenotypes. The empirical estimation of these thus consists of three steps. First, we perform a statistical (Bayesian) comparison of learning models [58]. For each subject, we fit trial-by-trial actions sequences $a_{1:60}$ with each learning model ($m \in \{$*BN, WSLS, RL, 0-ToM, Inf, 1-ToM, 2-ToM, 3-ToM*$\}$) using a variational Bayesian approach [59,60]. This eventually yields 8x48x4x2x2 = 6144 model evidences $p(a_{1:60}|m)$ (8 models, 48 participants, 4 opponent conditions, 2 framing conditions, 2 repetitions).

Second, we define the *repertoire's flexibility* $\hat{f}^{(1,2)}$ (between conditions 1 and 2) in terms of the posterior probability that a given participant employs different adaptation strategies across two conditions:

$$\hat{f}^{(1,2)} = p(m^{(1)} \neq m^{(2)} | a_{1:60}^{(1)}, a_{1:60}^{(2)}) = 1 - \sum_m p(m^{(1)} = m | a_{1:60}^{(1)}) p(m^{(2)} = m | a_{1:60}^{(2)}) \quad (8)$$

where $m^{(1)}$ (resp. $m^{(1)}$) is the participants' adaptation strategy in the first (resp. second) condition, $p(m^{(1)} = m|a^{(1)}_{1:60})$ (resp. $p(m^{(2)} = m|a^{(2)}_{1:60})$) is the probability that the participant had an adaptation strategy $m$ given his trial-by-trial choice sequence $a^{(1)}_{1:60}$ (resp. $a^{(2)}_{1:60}$) in condition 1 (resp. condition 2). Note that we measure the *repertoire's flexibility* $\hat{f}$ both across framings and across repetitions.

Third, we define the *repertoire's ToM-sophistication* $\hat{k}$ in terms of the expected depth of recursive belief update:

$$\hat{k} = E[k|a_{1:60}] = \sum_k k\, p(k|a_{1:60}) \tag{9}$$

where $p(k|a_{1:60}) = p(m = "k–ToM"|a_{1:60})$ is the posterior probability of model $k$-*ToM* given the participant's trial-by-trial choice sequence $a_{1:60}$. Note that we restrict the summation in Eq 9 to $k$-*ToM* models, because the depth $k$ of recursive beliefs is not defined for the other learning models. Note that we measure the *repertoire's ToM-sophistication* $\hat{k}$ in both framing conditions (social and non-social).

All statistical analyses were performed using the VBA toolbox [36], which contains the above eight learning/decision models as well as the bayesian statistical machinery required for model inversion.

## Supporting information

**S1 Text. This document provides supplementary information regarding: the experimental protocol (section 1), the credibility of the framing manipulation (section 2), the effect of motivational manipulations on the game's performance (section 3), differences in reaction times (section 4), model-free Volterra decompositions of trial-by-trial choice sequences (section 5), differences in adaptation strategies between AS and NT participants (section 6), the impact of the framing manipulation on the repertoire's flexibility (section 7), the statistical relationships between computational phenotypes (section 8), and their relationship with performance in the game (section 9).**
(DOCX)

## Acknowledgments

Authors thank Alexandra Duquette and Patricia Jelenic for contributing with data collection, and Pr. Laurent Mottron for enabling the recruitment of ASD patients.

## Author Contributions

**Conceptualization:** Baudouin Forgeot d'Arc, Marie Devaine, Jean Daunizeau.

**Data curation:** Marie Devaine.

**Formal analysis:** Marie Devaine, Jean Daunizeau.

**Funding acquisition:** Baudouin Forgeot d'Arc, Jean Daunizeau.

**Investigation:** Baudouin Forgeot d'Arc, Marie Devaine, Jean Daunizeau.

**Methodology:** Marie Devaine, Jean Daunizeau.

**Project administration:** Baudouin Forgeot d'Arc, Jean Daunizeau.

**Resources:** Baudouin Forgeot d'Arc, Marie Devaine, Jean Daunizeau.

**Software:** Marie Devaine, Jean Daunizeau.

**Supervision:** Jean Daunizeau.

**Validation:** Marie Devaine, Jean Daunizeau.

**Visualization:** Jean Daunizeau.

**Writing – original draft:** Baudouin Forgeot d'Arc, Jean Daunizeau.

**Writing – review & editing:** Baudouin Forgeot d'Arc, Jean Daunizeau.

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
