## [Decision Letter · Decision Letter 0]

9 Sep 2019

Dear Dr Daunizeau,

We would like to invite you to submit a revision of your manuscript to PLOS-CB, as the comments and suggestions of the two reviewers below would add greatly to the value of the paper. If I were to pick a theme which is most important to address, is the one of validity and subsequent diagnostic value of the method advocated here. As far as using this test in a clinical setting is concerned, the analysis that this manuscript present does not improve on the accuracy (hence scientific validity) of clinician diagnosis, and in fact this is impossible if the latter is taken as the measure by which the task-based diagnosis is judged. Of course this is most problematic if such tasks are seen as stand-alone assessments; They may indeed be very useful as screening tools (with the caveat that they then have to be tested against much more straightforward self- and collateral- reported questionnaires). Please read through the automated mail below that corresponds to our decision to ask you to submit a major revision of your manuscript, paying special attention to address all the reviewers' comments point-by-point, and rewriting the relevant sections of the manuscript to match.

                                                                   ~ ~ ~ ~ ~ ~ ~ ~

Thank you very much for submitting your manuscript 'Social behavioural adaptation in Autism' for review by PLOS Computational Biology. Your manuscript has been fully evaluated by the PLOS Computational Biology editorial team and in this case also by independent peer reviewers. The reviewers appreciated the attention to an important problem, but raised some substantial concerns about the manuscript as it currently stands. While your manuscript cannot be accepted in its present form, we are happy to consider a revised version in which the issues raised by the reviewers have been adequately addressed. We cannot, of course, promise publication at that time.

Sincerely,

Michael Moutoussis

Guest Editor

PLOS Computational Biology

Samuel Gershman

Deputy Editor

PLOS Computational Biology

[LINK]

Reviewer's Responses to Questions

**Comments to the Authors:**

Reviewer #1: In this excellent ms. the authors ambitiously suggest to "decompose reciprocial social interactions into their computational constituents". This goal is pursued by asking 24 adult autistic participants to 24 neurotypical (NT) participants engage in a "repeated dyadic competitive game against artificial agents with calibrated reciprocal adaptation capabilities".

More concretely, the game includes playing against a learning machine that algorithmically adapts to the player's behavior. To win, participants' must learn to anticipate their opponent's next choice and/or try to influence it. Importantly, a cover story was used to introduce a social condition, in which participants thought they were playing against a real person.

I am very impressed by goal of the study and the methodology used to assess cognitive processes that subtend behavioral performance. However, I am less impressed by the choice of the task, because it seems to be yet another example of a task with relatively low ecological validity. I would be more excited, if someone could demonstrate how performance on such a task relates to everyday life interaction difficulties that autistic persons experience. It is nice to show that mentalizing performance is suboptimal on this task, but what if mentalizing is not the key ingredient to successfully navigating the real social environment? Please discuss this particularly in the context that your analysis predict (only) 62% of the severity of social symptoms.

Other comments:

How do the authors explain that NT participants lose in the non-social condition, but retain earnings in the social condition? What is the role of motivation here?

The most interesting finding in my mind is that patients do better than NT in dealing with non-mentalizing agents. Could that be because this behavior is more 'autistic', more similar to patients' expectation? Bolis et al. have recently proposed the 'social interaction mismatch' hypothesis, which suggests that autistic persons find it easier to relate to other autistic persons (or agents). Please discuss.

With regard to the highly important topic of social reciprocity the authors do not discuss revent developments and relevant papers in social neuroscience (Schilbach et al. 2013, BBS; Schilbach 2016, Phil Trans Roy Soc; Redcay & Schilbach 2019, Nat Rev Neurosci) that have addressed this issue both conceptually and empirically.

Having said all of this, I think that the paper could make an important contribution, but should be revised to address the above described comments.

Reviewer #2: In this work, the authors aim to decompose and measure reciprocal social interactions to predict ASD diagnosis. The authors present behavioral results for ASD and TD individuals performing a competitive game against artificial agents with different levels of intelligence and theory of mind. The experimental design included four different AI agents, two framing conditions (social and non-social) and two financial incentives (high vs. low). The authors main finding is that only TD individuals changed their behavior in response to the social framing, showing superior performance of TD compared to ASD when the task was framed to make individuals believe that they were competing against somebody else. Authors used the financial incentives manipulation to suggest that motivational factors did not play a role in the after mentioned framing x group observed effect. Furthermore, the authors modeled several strategies (theory of mind, RL, random choice) to explain individuals' behavior, while also modeling the possibility that participants shifted between different strategies. Modeling result suggest lower flexibility and metalizing tendencies for ASD group. The authors report 79% diagnosis classification accuracy in a leave-on-out logistic regression, and suggest that the method and modelling approach may prove useful for novel diagnostic methods for autism.

First, allow me to thank you for the opportunity to review this paper. I believe the use of sophisticated AI agents to objectively measure human behavioral tendencies is promising and relevant. I would like to also note that the authors expertise in modeling is well observed, and the manuscript seem to include a rich and sophisticated modeling section that, in my opinion, fits well with the journal scope and aims. However, as noted below I have a few major concerns, the main of which regards the claim that these results are valuable diagnostically. I do hope the authors will find these comments helpful, and would be happy to see further versions in the future.

Major comments:

1. My main concern is the statement made by the authors claiming that the results can be taken as support for novel diagnosis metric. I appreciate the LOO CV result, and they are indeed impressive. However, it is my view that this reflects a large, and consistent effect within the current sample, and as such do not add much over the ANOVA results. The current study do not address within group variability or the issue of whether this method can be generalize outside the current sample (keeping in mind that ASD is a highly heterogeneous group). In that sense the CV results should be taken carefully, mostly as resonating the ANOVA results. Furthermore, the authors do not show whether symptoms severity, or specific clinical symptoms can/cannot be predicted using this approach, which might be very important. It might be also useful if the authors can find a way to shed some light on individuals that where miss classified. Did those individuals had stronger framing effect? Is there any reason to suggest they were miss-classified by the clinicians (which might actually mean this approach can add additional value to the current clinical approach)?

Second, the current test provides, like almost all measures in the field, an assessment of some symptoms that should be assumed to be predicted by ASD (not vice versa). I agree that a task-based objective measure could facilitate symptom-based diagnosis, which is currently done only by interviews and self-report. However, I think the authors need to specify how exactly this could be helpful. My understanding is that in the current sample the classification accuracy is not much worse than clinical measures. Given that task based estimates tend to be in general much less reliable compared to self-report measures (e.g., Enkavi et al., 2019), we can only assume that in the long run, it might be very difficult to justify the use of such approach as a clinical measure.

2. ASD individuals might have unpleasant memories from social interactions, more compared to TD individuals. Therefore, is it possible that for the ASD group the social version is less engaging, leading to worse performance compared to TD? I appreciate the fact that the authors manipulated reward magnitude, yet this might be somewhat unrelated since the claim here is that ASD might have been more anxious in the social condition, lowering their performance (What could really help in future studies is a social relevant clinical control group like individuals with social anxiety).

3. The authors use a correlational analysis to support the fact that the ASD group show no framing effect, which is essentially a null result. While the group x framing interaction might be enough here, I am not sure I understand why the correlational analysis in the SI, section 4 helps ith NHT issues regarding the simple effect of framing for the ASD group. Since the whole distribution can be shifted between conditions, the correlations can be high or low regardless of whether a simple framing effect is present within the ASD group. I would suggest that the authors use a Bayesian ANOVA, and report effect-size measures such as partial et-square (or any other) to demonstrate the main result – no framing effect for ASD compared to TD.

4. Is it possible that the k-ToM model fits the ASD better, thus leading to lower flexibility? If this is the case, is it possible that ASD find it harder to play the task due to the fact that the AI agent is more suitable opponent for their strategy? Finally, could this mean that they do not show framing effect for high k opponents, due to the fact that the game is (much) more difficult for them, not leaving much room for improvement?

5. Given the 1.3 response deadline, is it possible that the ASD (which can show prolonged RTs) had to lower their response threshold, and suffered from less optimal speed-accuracy trade-off point?

6. I think the results regarding flexibility are not explained with enough data. Is there any way the authors can outline the fit of the different models, and give the reader some understanding of which models where used by which groups? What is the relationship between flexibility parameter and overall model fit?

Minor comments:

- The author describe former ToM tests as having 'unreliable results and poor psychometric properties' (bottom, pg3). I think it would be great if the authors could specific more exact reliability/validity estimates, and show where possible how the current test differ, if they are to keep the claim that this method is proved to be clinically valuable.

- I understand the need to keep some results out of the main text, but please clearly note the full design in the main text (including the reward manipulation) so the reader will know what to except.

- Fig 1 – I'm not sure I understood why use a net rate rather than the more straight forward accuracy rate?

- The authors report in the first paragraph of the discussion four analyses to support their claims yet three of which are tacked in the SI, which seems to be written with less care. Is there any way to improve the presentation?

- The authors suggest that a completely random strategy is "embarrassingly simple". I would argue that asking a human individual to act in a perfectly random manner is rather difficult, and against some human opponents could be useful to shift to. In that sense, it is interesting to see what role this strategy played in the individuals data.

Typos:

- SI section 4:" whether there are there",

- "Figure A2 below show the correlation between" I think you meant Fig A3

- " (Fig A3, right inset) analyses show a significant correlation between framings for the AS group" I think you meant between repetitions.

- Please carefully proof the SI as it seems to include valuable parts of the study.

**Have all data underlying the figures and results presented in the manuscript been provided?**

Reviewer #1: Yes

Reviewer #2: Yes

PLOS authors have the option to publish the peer review history of their article (what does this mean?). If published, this will include your full peer review and any attached files.

Reviewer #1: No

Reviewer #2: No

---

## [Decision Letter · Decision Letter 1]

31 Dec 2019

Dear Dr Daunizeau,

Thank you very much for submitting your manuscript, 'Social behavioural adaptation in Autism', to PLOS Computational Biology. As with all papers submitted to the journal, yours was fully evaluated by the PLOS Computational Biology editorial team, and in this case, by independent peer reviewers. The reviewers appreciated the attention to an important topic but identified an aspect of the manuscript that should still be addressed.

We feel that your article is very near to a standard where it can be accepted for publication in PLOS-CB, and it will not need to be returned to the reviewers.

I would like to comment on the points still raised by Reviewer 2, and ask that you kindly respond in the Discussion to their first point, which regards response times. I think that Reviewer 2's second point, although of some validity, will not be a serious concern for our readership. The term 'clinical utility' may be changed, for example, to 'clinical relevance' at the autor's discertion, but otherwise the Discussion (and slightly less so the Introduction) now make it clear that the paper makes an incremental contribution towards 'bedside' tests for AS, rather than any exaggerated claims. However, I would kindly ask the authors to address the point regarding response times. I feel that Reviewer 2's hypothesis is rather too elaborate, requiring that AS participatns were slower in general, but not so slow as to make more mistakes in the nonsocial condition where they performed better, yet too slow for the social condition. Reviewer 2 does have a point, that high-ToM may be too costly, in terms of response time, for AS participants rather than then being altogether unable to consider it. Therefore, when they attempt such a high-ToM strategy their performance deteriorates and may resemble 'Influence' decision-making. However I feel that RT analysis is outside the scope of this paper and should not delay its publication. I would therefore suggest that the authors address Reviewer 2's suggestion that RT analysis would make conclusions more compelling by discussing it in an additional paragraph on future research directions. Response time analysis - among other methods - may indeed move understanding from the algorithmic level of this paper to one incoroporating neural implementational features (such as accummulator processes that generate response times) and this may be addressed in future work.

We would therefore like to ask you to modify the manuscript according to the review recommendations. Your revisions should address the specific points made by each reviewer and we encourage you to respond to particular issues Please note while forming your response, if your article is accepted, you may have the opportunity to make the peer review history publicly available. The record will include editor decision letters (with reviews) and your responses to reviewer comments. If eligible, we will contact you to opt in or out.raised.

- Supporting Information uploaded as separate files, titled 'Dataset', 'Figure', 'Table', 'Text', 'Protocol', 'Audio', or 'Video'.

We hope to receive your revised manuscript within the next 30 days. If you anticipate any delay in its return, we ask that you let us know the expected resubmission date by email at ploscompbiol@plos.org.

Sincerely,

Michael Moutoussis

Guest Editor

PLOS Computational Biology

Samuel Gershman

Deputy Editor

PLOS Computational Biology

[LINK]

Reviewer's Responses to Questions

**Comments to the Authors:**

Reviewer #1: The authors have satisfactorily addressed my previous comments in their revision.

Reviewer #2: Dear Authors,

Thank you for your answers - much appreciated. Please find below two followup issues.

Many thanks.

R2.

1. Given the 1.3 response deadline, is it possible that the ASD (which can show prolonged RTs) had to lower their response threshold, and suffered from less optimal speed-accuracy trade-off point?

We think this is unlikely, for two reasons. First, AS and NT participants show no significant difference in IQ, including measures of processing speed. Second, AS participants perform significantly better than controls against non-mentalizing agents, in both social and non-social framing conditions…

Thank you for your reply, but I am not convinced. A response deadline of 1300ms is very strict and could majorly affect decision threshold. ASD are known to show prolonged RTs and increased RTV (mostly a higher rate of very slow RTs, and mostly due to co-occurrence with ADHD symptoms). I would encourage the authors to give this more thought. For example, it might be that with higher mentalizing agents (RB to 2-TOM) all individuals required more time to make a decision. Given that ASD are assumed to be slower in general, the strict RT deadline might render the high TOM conditions much more difficult for them, which can then reduce their ability to show sensitivity to the framing effect. An RT analysis, given this strict response deadline would thus make your mode-free analysis much more compelling. PS pencil-paper test is not enough to indicate no RT differences between the groups as it does not have the required sensitivity.

2. In my previous comments, I suggested that my main concern is the statement made by the authors claiming that the results can be taken as support for novel diagnosis metric. I appreciate the authors changes in the study, but it is my personal view that this paper still creates strong expectations for the reader regarding clinical relevance. For example, the authors note in the first paragraph of the intro that “This work evaluates the clinical utility of a computational decomposition...”, and mention the need for a ‘true test’ for ASD.

**Have all data underlying the figures and results presented in the manuscript been provided?**

Reviewer #1: Yes

Reviewer #2: No: The authors note they intend to make raw data accessible online upon acceptance.

PLOS authors have the option to publish the peer review history of their article (what does this mean?). If published, this will include your full peer review and any attached files.

Reviewer #1: No

Reviewer #2: No

---

## [Editor Report · Decision Letter 2]

30 Jan 2020

Dear Dr. Daunizeau,

We are pleased to inform you that your manuscript 'Social behavioural adaptation in Autism' has been provisionally accepted for publication in PLOS Computational Biology.

Before your manuscript can be formally accepted you will need to complete some formatting changes, which you will receive in a follow up email. A member of our team will be in touch within two working days with a set of requests.

Best regards,

Michael Moutoussis

Guest Editor

PLOS Computational Biology

Samuel Gershman

Deputy Editor

PLOS Computational Biology

---

## [Editor Report · Acceptance letter]

6 Mar 2020

PCOMPBIOL-D-19-01204R2 

Social behavioural adaptation in Autism

Dear Dr Daunizeau,

I am pleased to inform you that your manuscript has been formally accepted for publication in PLOS Computational Biology. Your manuscript is now with our production department and you will be notified of the publication date in due course.

With kind regards,

Bailey Hanna
